# Analyses of cancer incidence and other morbidities in gamma irradiated B6CF1 mice

**Alia Zander, Tatjana Paunesku, Gayle E. Woloschak**◉*

Feinberg School of Medicine, Radiation Oncology, Northwestern University, Chicago, IL, United States of America

* g-woloschak@northwestern.edu

**Data Availability Statement:** The data underlying the results presented in the study are available from http://janus.northwestern.edu/janus2/index.php.

## Abstract

With increasing medical radiation exposures, it is important to understand how different modes of delivery of ionizing radiation as well as total doses of exposure impact health outcomes. Our lab studied the risks associated with ionizing radiation by analyzing the Northwestern University Radiation Archive for animals (NURA). NURA contains detailed data from a series of 10 individual neutron and gamma irradiation experiments conducted on over 50,000 mice. Rigorous statistical testing on control mice from all Janus experiments enabled us to select studies that could be compared to one another and uncover unexpected differences among the controls as well as experimental animals. For controls, mice sham irradiated with 300 fractions died significantly earlier than those with fewer sham fractions and were excluded from the pooled dataset. Using the integrated dataset of gamma irradiated and control mice, we found that fractionation significantly decreased the death hazard for animals dying of lymphomas, tumors, non-tumors, and unknown causes. Gender differences in frequencies of causes of death were identified irrespective of irradiation and dose fractionation, with female mice being at a greater risk for all causes of death, except for lung tumors. Irradiated and control male mice were at a significantly greater risk for lung tumors, the opposite from observations noted in humans. Additionally, we discovered that lymphoma deaths can occur quickly after exposures to high doses of gamma rays. This study systematically cross-compared outcomes of different modes of fractionation evaluated across different Janus experiments and across a wide span of total doses. It demonstrates that protraction modulated survival and disease status differently based on the total dose, cause of death, and sex of an animal. This novel method for analyzing the Janus datasets will lead to insightful new mechanistic hypotheses and research in the fields of radiation biology and protection.

## Introduction

Ionizing radiation is an unavoidable risk in daily life and understanding its biological impacts is important for setting radiation protection standards. Approximately half of humankind's cumulative annual radiation exposure comes from natural sources, such as cosmic radiation and soil; the other half is derived from human-made sources including medical procedures

**Funding:** National Institute of Health grants R01OH010469 and RO1CA221150 were both awarded to GEW.

**Competing interests:** The corresponding author of this paper, Gayle E. Woloschak, is a section editor for PLOS ONE. This does not alter our adherence to PLOS ONE policies on sharing data and materials.

and nuclear medicine [1]. Most of the general population receives low dose chronic ionizing radiation exposures, accumulating to a few hundred mSv over a lifetime [2].

Calculating the risks associated with these chronic exposures is challenging because the overall effect of these lower dose/dose-rate exposures is small compared to the baseline risk of the same diseases. There are several helpful data sources that researchers have utilized to help quantify these risks including radiation therapy studies, atomic bomb survivor data, and other epidemiological studies from workers in the field and nuclear disasters. The radiation doses given to patients for radiation therapy are much larger than standard exposure and are only used on a small segment of a patient, not whole-body exposures. The major source of data on whole-body human exposures to gamma radiation is the Life Span Study (LSS) cohort that includes over 120,000 survivors of the atomic bombing in 1945 [3–6]. While these data have been a remarkable resource for epidemiological studies determining risks associated with acute exposures [5, 7–13], extrapolation of health risks to humans exposed intermittently to lower doses of radiation remains uncertain. Different mathematical modelling approaches have been used over the past 50 years to extrapolate health risks but they were met with variable enthusiasm from the scientific community [1, 14–18]. Ultimately, epidemiological studies are affected by confounding factors and uncertainties making well-controlled animal studies a valuable resource to supplement conclusions from human studies [19].

We utilized the Northwestern University Radiation Archive for animals (NURA), a source of irradiated animal data documenting findings from Janus studies conducted between 1972 and 1989 at Argonne National Laboratory (ANL). Ten large volume experiments with B6CF1 mice were designed to determine the effects of acute and fractionated whole-body radiation on survival and causes of death [20–22]. Over 50,000 male and female mice were exposed to acute or fractionated neutrons or gamma rays, at ages between 90–200 days or more than 500 days. Moribund animals were sacrificed and necropsy results were recorded. This experiment was one of the of the largest ever conducted in the USA; at the conclusion of these studies the Janus irradiator and other irradiation facilities at ANL were dismantled making it unlikely that experiments of this scope will be repeated. Numerous studies used the NURA (also known as Janus) database. In most cases, different Janus experiments were used separately [17, 20, 23–28] or else combined all together into a single dataset [14]. In this study, however, many but not all Janus experiments were combined into a dataset. The selection process for inclusion was based on comparability of control animal datasets from sham irradiation conditions in different Janus experiments.

Similar studies on other strains of mice were conducted in Europe [29] and more recent work conducted at the Institute of Environmental Sciences (IES) in Japan explored chronic exposures in a closely related animal strain. Tanaka and others compared findings on 4,000 B6C3F1 mice of both genders that were irradiated for 22h daily for approximately 400 days using low dose rate gamma rays with accumulating total doses of 0, 20, 400, or 8000 mGy [30]. As the experiments performed at IES are similar to Janus experiments with regard to number of animals and total doses, we performed a side by side comparison of cancer incidence to determine biological similarities of these findings.

We examined whether fractionation, age at which a mouse was first irradiated, and gender modulated the overall death hazard and frequency for specific causes of death in gamma irradiated mice. Our approach included use of general Cox proportional hazards models, cumulative incidence function models, and cause specific hazards models [31–33]. We found that the two approaches to represent and evaluate competing risks from the same data complement each other and improve insight into effects of gamma ray fractionation. While this work cannot be directly translated into recommendations for radiation protection policies, it brings to our attention the fact that it should be possible to standardize comparisons between different

types of fractionated exposures and perhaps fractionated versus chronic radiation exposures across a large span of total doses. A single mathematical formula cannot be used universally for conversion between any two possible radiation exposure scenarios, but it is possible that the growth of machine learning and artificial intelligence techniques will permit us to craft realistic approaches to predict changes in health complication spectra from one irradiation exposure to another, possibly even among different species. As we prepare for this future, it is necessary to ensure that we preserve radiation data archives with as much granularity as possible. This study is a prime example of utilizing archives by analyzing data in a new light to further augment our understanding of radiation biology.

## Methods

### Data selection—NURA

Argonne National Laboratory conducted a series of 10 large scale ionizing radiation lifespan studies on rodents between 1972 and 1989. These studies are now part of the NURA archive housed by the Woloschak laboratory and posted on the web, allowing access to all who are interested in this dataset [14, 21, 22, 34]. Records list individual mouse information with the type of radiation, total dose, dose rate, fractionation schedule, age first irradiated, age at death, cause of death and, in many cases, detailed pathology analyses. All animals received whole body external beam ionizing radiation from cobalt 60 gamma rays or neutrons [20–22, 35]. Most control mice were sham irradiated–transported from their housing location to the room with the irradiator turned off. Background radiation levels in animal housing rooms were closely monitored. Mice listed in S1 Table in S2 File were censored due to early exit from the study because of causes unrelated to the experimental plan.

To ensure that any significant changes among different groups of mice were due to modulations in radiation exposure and not due to changes in baseline survival, we filtered out groups of mice that showed statistically significant survival probability differences. S2 Table in S2 File details groups of mice that exhibited sufficient survival variation in control animals to warrant removal of the specific data from our analysis. For this analysis, we focused on gamma irradiated mice. Neutron irradiated mice were studied in a separate analysis. Breeder mice were not used as controls for any of these analyses because of their unique housing conditions. Only *Mus musculus* species B6CF1 strain mice were used for this work; different species such as *Peromyscus leucopus* (white-footed deer mouse) were excluded from this study because of the species to species differences between the controls and in response to radiation [27]. Similarly, mice treated with radioprotectors [28] were also removed from this study. As a result of data refinement, only two of the ten experiments were completely removed from this study.

The predicted model output graphs from Cox Proportional Hazards (PH) analyses of sham irradiated control mice are shown in S1 Fig in S2 File along with parameter estimates and p-values. Each overall model was significant due to sex, but no other covariates were significant in their respective models. Additionally, Kaplan Meier (KM) curves [36] in S1e–S1h Fig in S2 File validate the proportional hazards assumption for our model. To further validate our model, we used robustness tests, making small modifications to each variable in our models, shown in S3 Table in S2 File and described in more detail in the Supplementary methods in S1 File.

### Survival analysis

Kaplan-Meier (KM) curves were used for categorical univariate survival analysis using the "survfit" function in the survival package in R [37, 38]. Cox proportional hazard (PH) models were used to analyze survival over time with multivariate models that included a mixture of

categorical and quantitative predictor variables and interactions between variables [31]. The main models used for Cox PH with sham irradiated mice and shown in S1 Fig in S2 File are as follows:

$$\lambda(t) = \lambda_0(t)e^{(\beta_1 sex + \beta_2 experiment)} \qquad (A)$$

$$\lambda(t) = \lambda_0(t)e^{(\beta_1 sex + \beta_2 fractions)} \qquad (B)$$

$$\lambda(t) = \lambda_0(t)e^{(\beta_1 sex + \beta_2 first\ irrad)} \qquad (C)$$

Our main Cox PH model for gamma irradiated mice is as follows:

$$\lambda(t) = \lambda_0(t)e^{(\beta_1 sex + \beta_2 first\ irrad + \beta_3 total\ dose + \beta_4 fractions + \beta_5 total\ dose:fractions)},$$

where $\lambda(t)$ is the hazard function based on our set of covariates including sex, age first irradiated, total dose, number of fractions, and the interaction between total dose and fractions; β is a vector of their corresponding coefficients, and $\lambda_0(t)$ is the baseline hazard. All Cox PH models were performed using the coxph function from the survival package in R [38].

## Competing risks analysis

A competing risk is anything that decreases the likelihood of an outcome of interest. When looking at specific causes of death, all other causes of death fall into this category. For the competing risks analysis, we examined crude incidences, cause-specific hazard models, and cumulative incidence function (CIF) regression models [39]. In the absence of competing risks, the cumulative incidence of events over time can be measured using one minus the Kaplan-Meier estimate of the survival function. In the presence of competing risks, the KM method results in upward biases for the CIF [32]. We used the "cuminc" function from the "cmprsk" package in R to investigate crude, nonparametric incidences in the presence of competing risks [40].

For multi-variable regression analyses in the presence of competing risks, we used both cause specific hazards and CIF models. The cause specific hazards were estimated using the "coxph" function in R [38]. All causes of death, excluding the event of interest, were censored. Concretely: $\lambda_k(t) = \lambda_{0k}(t)e^{(\beta_1 sex + \beta_2 first\ irrad + \beta_3 total\ dose + \beta_4 fractions + \beta_5 total\ dose:fractions)}$, subset data for age first irradiated $< 500$ days.

where $\lambda_k(t)$ is the hazard function for the $k^{th}$ cause of death. The cause specific hazards method is used to determine the effect that covariates have on all event free subjects. The cumulative incidence function describes the overall probability of a particular outcome and does not depend on a subject being event free [32, 33, 41, 42]. Concretely: $\lambda_k^*(t) = \lambda_{0k}(t)e^{(\beta_1 sex + \beta_2 firsr\ irrad + \beta_3 total\ dose + \beta_4 fractions + \beta_5 total\ dose:fractions)}$, subset data for age first irradiated $< 500$ days.

where $\lambda_k^*(t)$ is the subdistribution hazard function for the $k^{th}$ cause of death. Cumulative incidence hazards were estimated using the "crr" function in the "cmprsk" package in R [40].

## Cause of death groupings

We used data downloaded from the Janus website listed as "Grouped Macros," which includes all pathologies found in animals at the time of death and categorizes them as lethal (L), contributory (C), or non-contributory (N). For the purposes of our investigation, we only examined lethal diseases. To make the data more robust for analyses, we grouped causes of death (CODs) into lymphomas, tumors other than lymphomas–referred to as tumors (sometimes separating them into lung tumors and tumors or no lung tumors), non-tumors, or causes of

death unknown (CDU). Specific analyses of diseases affecting the liver, lung, kidney, and vascular system for a subset of these data were previously conducted [43].

### Data reformatting for comparisons with IES data

Studies at IES involved low dose rate gamma irradiations of specific-pathogen free (SPF) B6C3F1 mice, F1 progeny of C57BL/6J females and C3H/HeJ males. The B6CF1 mice, F1 progeny of C57BL/6J females and BALB/cJ males, were used during the Janus experiments. Both strains are F1 hybrids that share the same maternal strain C57BL/6. The differences in disease incidence between the control animals point out that only some of these disease "endpoints" are appropriate for direct comparisons between strains when different "test conditions" are being evaluated.

During the IES studies, (SPF) B6C3F1 mice were irradiated with low dose rate $^{137}$Cs gamma rays for 22 hours a day, beginning the irradiations with acclimated 8-week old animals in a sterile environment. Chronic exposures of 0.05, 1.1 or 21 mGy/day continued for 400 days leading to total doses of 2, 40, or 800 cGy. Similar to the Janus experiments, many mice from the IES experiments were allowed to live out their entire lifespan, and each mouse was assigned a single cause of death [30, 44]. These data were included in this study because of their similarity to the experiments carried out on B6CF1 mice during the NURA experiments.

To compare the Janus data included in this study with the results from the IES studies [30], we grouped Janus CODs to match IES CODs as closely as possible (S4 Table in S2 File). The total doses used for Janus experiments spanned a larger range than those used for IES experiments. To make the comparisons more meaningful, we limited the Janus data used for this particular analysis to a subset of conditions that closely matched the IES dataset conditions. S5 Table in S2 File provides information about the Janus data included in this comparison.

### Tools and scripts

Files stored on github: https://github.com/aliazander

## Results

### Mice sham irradiated with 300 fractions had decreased survival

Control mice that received 300 fractions (5 fractions/week) of sham irradiation died significantly earlier than control animals that received fewer than 300 fractions of sham irradiation. This was evident in Cox PH models using sex and fractions as independent variables. This result held true with fractions treated as a continuous variable (Fig 1A and 1B, p-value <0.001) and as a categorical variable (S2A and S2B Fig in S2 File, p-value <0.001). KM curves showed a very similar trend to the predicted outcomes from the Cox PH models and validated the proportional hazards assumption of the Cox PH model (S2C Fig in S2 File). Because general stress is the only probable cause for the increased death hazard observed in this group of animals and irradiated animals exposed to 300 fractions most likely experienced the same stress, we excluded mice exposed to 300 fractions from our main analysis. These mice were included as part our robustness testing.

### Mice sham irradiated with 300 fractions had significant changes in causes of death compared to other control mice

The decreased survival in mice that received 300 fractions compared to fewer than 300 fractions during their sham irradiations led us to investigate the specific causes of death for these two groups of mice. All animals sham irradiated with 300 fractions were male. Using non-

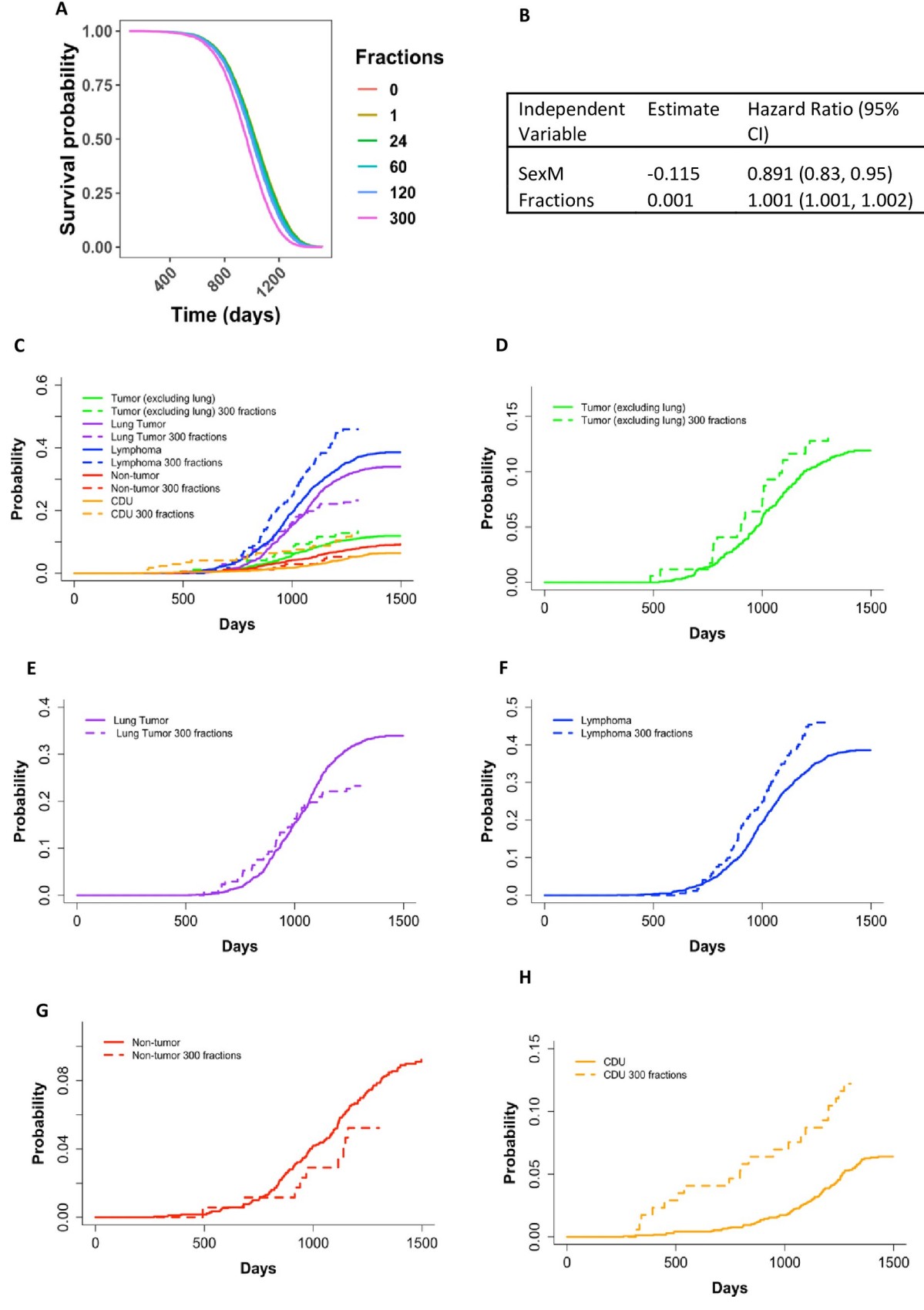

**Fig 1.** Survival probability output from Cox PH model for control mice with sex and the number of fractions as a continuous variable as independent variables **(A)**. The predicted outcome shown in **(A)** are for male mice and output values shown in (B), p-value for fractions <0.001. **(C)** Non-parametric cumulative incidence over time for specific causes of death grouped as tumors (excluding lung tumors), lung tumors, lymphomas, non-tumors, and CDUs. Dashed lines represent mice that received their sham irradiations in 300 fractions and solid lines represent mice that received their sham irradiations in fewer than 300 fractions. P-values for the differences of incidences between mice that received their sham irradiations in 300 fractions vs. mice that received their sham irradiations in fewer than 300 fractions: Tumor– 0.348, Lung Tumor– 0.038, Lymphoma—0.017, Non-tumor 0.222, CDU– 0.0002. For closer examination, we plotted each cause of death CIF individually–**(D)** tumors (excluding lung tumors), **(E)** lung tumors, **(F)** lymphomas, **(G)** non-tumors, and **(H)** CDUs.

parametric CIF, we found significant increases in lymphoma and CDU incidences and a significant decrease in lung cancer incidences in mice that received 300 sham irradiation fractions compared to all other sham irradiated male mice (Fig 1). Additionally, we examined how the number of fractions impacted survival probability over time through KM curves for each COD (S2D–S2H Fig in S2 File) and closer examination of the CIF curves in Fig 1C by individually plotting each COD (Fig 1D–1H). The initial onset for lung tumor deaths (Fig 1E; S2E Fig in S2 File, p = 0.04), CDU deaths (Fig 1H; S2H Fig in S2 File, p <0.001), and lymphomas (Fig 1F; S2E Fig in S2 File, p = 0.02) was earlier when mice received sham irradiations in 300 fractions, but there did not appear to be a difference for tumors (excluding lung tumors) (Fig 1E; S2C Fig in S2 File, p = 0.35) or non-tumors deaths (Fig 1G; S2G Fig in S2 File, p = 0.22). The KM curves also supported these findings (S2C–S2H Fig in S2 File).

## Aged mice were excluded from our analysis because of uneven experimental conditions

All control and gamma irradiated mice selected for this study (S2 Table in S2 File) are represented in a box and whisker plot of age at death versus total dose with colors indicating the number of fractions used. Gamma irradiated mice received 21.57 to 4901 cGy (Fig 2A), with the maximal dose for acute exposures limited to 546 cGy. Because the $LD_{50/30}$ for B6CF1 mice at 110 days of age is approximately 7 Gy [45], this maximum acute dose ensured that animals did not die from acute radiation syndromes. Fig 2A shows that fractionation had a larger impact on age at death as total doses increased. According to the Janus documentation, [21] the age first irradiated for mice was intended to be 100 days +/- 15 days, with a small subset of mice acutely irradiated at 500 days in order to investigate how age first irradiated impacted survival. Plotting the frequency of each age first irradiated, we found that the majority of mice were first irradiated within the expected range and there was a small group of mice that were first irradiated over 500 days old (Fig 2C). Fig 2B represents age at death against total dose with dark purple bars representing the aged mice and the light purple bars representing mice irradiated closer to 100 days of age. Due to the low sample size for aged mice, the large amount of leverage they would have on the overall model, the absence of sham irradiated aged mice, and the lack of direct dose comparisons with younger mice, we excluded these 560 mice from further analysis. This resulted in 11,618 total mice for the analysis on gamma irradiated mice.

## Fractionation increased the overall survival probability in mice exposed to gamma rays

To determine how fractionation impacts survival, we used a Cox PH model with age at death as the time scale and sex, age first irradiated, total dose, fractions, and the interaction between total dose and fractions as independent variables. All independent variables were significant in the model, except for age first irradiated (Table 1). Given the small range of ages first irradiated included in our sample, this was an expected result (p = 0.14). The main effect of fractions resulted in a positive coefficient from our model output, which corresponds to an increase in

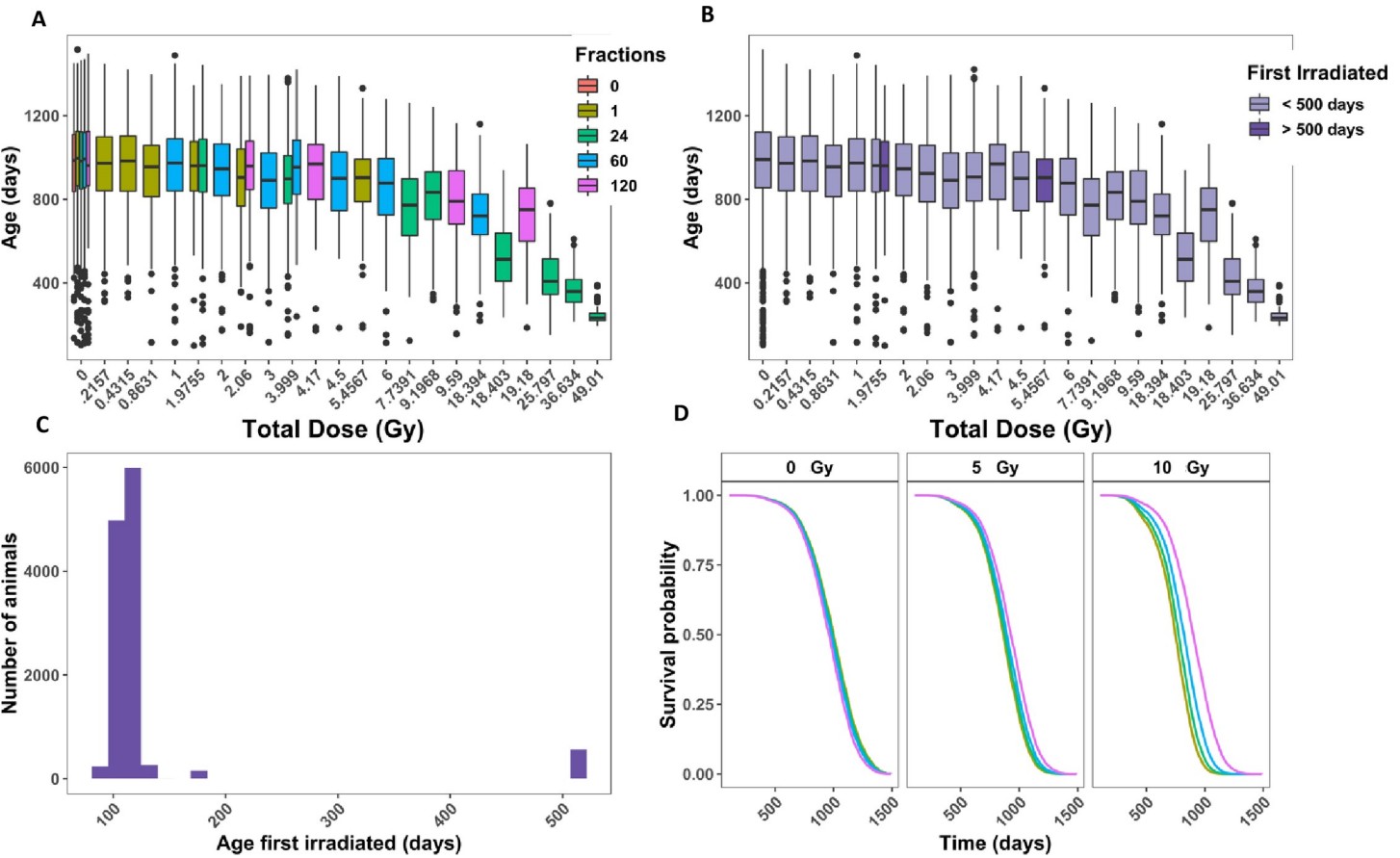

**Fig 2. Analysis of animals filtered in Table 1 that were controls or gamma irradiated. (A)** Box plot of age at death in days versus total dose in Gy. Colors indicate the number of fractions. **(B)** Histogram of the total number of animals versus age first irradiated in days. **(C)** Age at death in days versus total dose in Gy. Colors indicate whether a mouse was first irradiated before or after 500 days. **(D)** Representative graphs from Cox PH model output with age at death as the time scale and sex (p < 0.001), age first irradiated (p = 0.14), total dose (p <0.001), fractions (p = 0.001), and the interaction between total dose and fractions (p <0.001) as independent variables. The predicted outcomes shown are for female mice first irradiated at 120 days.

hazard (p = 0.001). This result is the outcome of the difference in total doses of exposure for acute and fractionated radiation regimens. The maximum acute exposure was 5.4Gy, but the total doses for fractionated exposures reached as much as 49Gy, causing the interaction between total dose and the number of fractions to be the most relevant for determining the role of fractionation. The interaction term between fractions and total dose was highly significant and its interpretation is best understood by graphical representation of the model's predicted outcome (Fig 2D, p <0.001). As the total dose increased, the beneficial effect of

**Table 1. Parameter estimates, hazard ratios with 95% confidence interval, and p-values for main Cox proportional hazards model in Fig 2D.**

| Variable | Estimate | Hazard Ratio (95% CI) | P-value |
|---|---|---|---|
| sexM | -0.17 | 0.84 (0.80, 0.89) | **<0.001** |
| Fractions | 0.002 | 1.00 (1.001, 1.003) | **0.001** |
| Total dose | 16.61 | 1.64E7 (8.2E6, 3.3E7) | **<0.001** |
| First irrad | -0.002 | 0.998 (0.996 1.00) | 0.137 |
| Fractions:Total dose | -0.109 | 0.89664 (0.88, 0.91) | **<0.001** |

fractionation became more pronounced. This result was consistent throughout a series of robustness tests (S3 Fig in S2 File). Notably, gamma irradiated mice that received their total doses in 300 fractions had a decrease in the death hazard compared to mice that received acute exposures, even with the added stress that caused control mice to die significantly earlier (S3I–S3L Fig in S2 File). When adding a new interaction term between sex and total dose, we found that males as the total dose increased, the decreased death hazard in males was more pronounced (S3M and S3N Fig in S2 File). We used KM curves to validate the proportional hazards assumption in our model and found parallel survival curves between groups based on sex, number of fractions, age first irradiated, and total dose (S4 Fig in S2 File).

## Fractionation significantly decreased the death hazard for mice dying from lymphomas, tumors, non-tumor, and causes of death unknown in gamma irradiated mice

When analyzing specific causes of death, it is important to consider the effects of competing risks. Cause specific hazards models are one type of competing risk model and their parameter estimates can be interpreted as the hazards for the specific event of interest. The cause specific hazards models for tumors, lymphomas, non-tumors, and CDUs all showed a similar trend (Fig 3A–3D)—as the dose increased, there was an increased rescue effect from fractionation, and more fractions corresponded to less hazard. Estimated model parameters showed that the interaction term between total dose and number of fractions was significant for all four

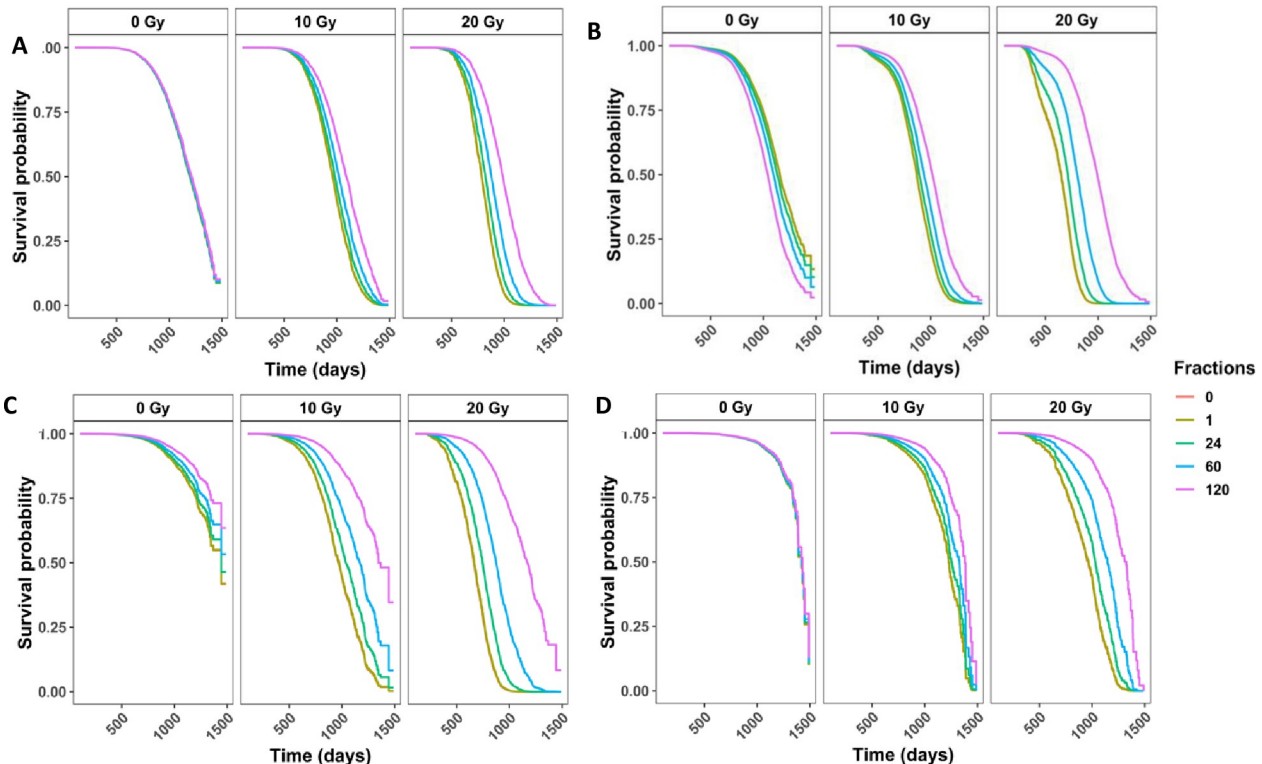

**Fig 3. Competing risks models for specific causes of death in gamma irradiated mice with age as a time scale and sex, age first irradiated, total dose, fractions, and the interaction between total dose and fractions as independent variables.** Survival curves for cause of death being **(A)** any solid tumors, **(B)** lymphomas, **(C)** non-tumors, and **(D)** cause of death unknown. Model estimates, confidence intervals and p-values are listed in the corresponding Table 2. All four models have a significant interaction term between total dose and the number of fractions (tumor p = 0.001, lymphoma p<0.001, non-tumor p <0.001, CDU p <0.001). The graphs represent predicted outcomes for female mice first irradiated at 120 days.

**Table 2. Competing risks model output for cause specific hazards and subdistribution hazards.**

| | | Cause Specific Hazards | | | Subdistribution Hazards | | |
|---|---|---|---|---|---|---|---|
| COD | Independent Variable | Estimate | Hazard Ratio (95% CI) | P-value | Estimate | Hazard Ratio (95% CI) | P-value |
| Tumors | sexM | 0.195 | 1.21 (1.11, 1.33) | **<0.001** | 0.453 | 1.573 (1.446, 1.711) | **<0.001** |
| | Fractions | -0.001 | 0.9994 (0.998, 1.001) | 0.535 | -0.003 | 0.997 (0.995, 0.999) | **0.002** |
| | Total dose | 0.123 | 1.13 (1.11, 1.15) | **<0.001** | -0.063 | 0.939 (0.93, 0.948) | **<0.001** |
| | First irrad | -0.0001 | 0.9992 (0.996, 1.003) | 0.859 | 0 | 1 (1, 1) | 0.71 |
| | Fractions: Total dose | -0.001 | 0.9994 (0.9991, 0.9998) | **0.001** | 0.001 | 1.001 (1, 1.001) | **<0.001** |
| Lymphoma | sexM | -0.37 | 0.69 (0.64, 0.75) | **<0.001** | -0.259 | 0.772 (0.72, 0.83) | **<0.001** |
| | Fractions | 0.005 | 1.005 (1.004, 1.007) | **<0.001** | 0.004 | 1.004 (1.002, 1.005) | **<0.001** |
| | Total dose | 0.166 | 1.18 (1.17, 1.19) | **<0.001** | 0.012 | 1.012 (1.004, 1.021) | **0.007** |
| | First irrad | -0.003 | 0.997 (0.993, 1.0001) | 0.058 | 0 | 1 (0.999, 1) | **.014** |
| | Fractions: Total dose | -0.001 | 0.999 (0.998, 0.999) | **<0.001** | 0 | 1 (1, 1) | .12 |
| Non-tumors | sexM | -0.50 | 0.61 (0.53, 0.69) | **<0.001** | -0.265 | 0.767 (0.676, 0.87) | **<0.001** |
| | Fractions | -0.005 | 0.994 (0.991, 0.998) | **0.001** | -0.007 | 0.993 (0.99, 0.997) | **<0.001** |
| | Total dose | 0.191 | 1.21 (1.19, 1.23) | **<0.001** | 0.07 | 1.073 (1.06, 1.086) | **<0.001** |
| | First irrad | -0.006 | 0.994, (0.988, 0.9999) | **0.009** | 0.001 | 1.001 (1.000, 1.001) | **<0.001** |
| | Fractions: Total dose | -0.001 | 0.999 (0.9990, 0.9995) | **<0.001** | 0 | 1 (1, 1) | .84 |
| CDU | sexM | -0.14 | 0.87 (0.71, 1.07) | 0.182 | -0.046 | 0.955 (0.79, 1.16) | 0.64 |
| | Fractions | -0.001 | 0.999 (0.995, 1.003) | 0.667 | -0.002 | 0.998 (0.994, 1.002) | 0.34 |
| | Total dose | 0.154 | 1.166 (1.14, 1.20) | **<0.001** | 0.002 | 1.002 (0.985, 1.02) | 0.82 |
| | First irrad | 0.001 | 1.001 (0.993, 1.01) | 0.782 | 3.96E-04 | 1.00 (1.00, 1.00) | 0.37 |
| | Fractions: Total dose | -0.001 | 0.999 (0.999, 0.999) | **0.011** | 0 | 1 (1, 1.001) | 0.54 |
| Lung tumors | sexM | 0.89 | 2.44 (2.12, 2.79) | **<0.001** | 1.19 | 3.30 (2.9, 3.8) | **<0.001** |
| | Fractions | -0 | 0.9999 (0.997, 1.002) | 0.944 | -2.90E-03 | 0.997 (0.995, 0.999) | **0.0097** |
| | Total dose | 0.11 | 1.12 (1.09, 1.14) | **<0.001** | -0.066 | 0.936 (0.925, 0.948) | **<0.001** |
| | First irrad | -0.004 | 0.996 (0.991, 1.001) | 0.130 | 0.001 | 1.001 (1, 1.001) | **.029** |
| | Fractions: Total dose | 0 | 0.9996 (0.999, 1.000) | 0.079 | 0.001 | 1.001 (1, 1.001) | **<0.001** |
| Tumors (excluding lung tumors) | sexM | -0.469 | 0.63 (0.55, 0.71) | **<0.001** | -0.38 | 0.684 (0.606, 0.773) | **<0.001** |
| | Fractions | -0.001 | 0.999 (0.996, 1.001) | 0.348 | -0.003 | 0.997 (0.995, 1) | **0.021** |
| | Total dose | 0.136 | 1.15 (1.12, 1.17) | **<0.001** | -0.057 | 0.944 (0.931, 0.958) | **<0.001** |
| | First irrad | 0.003 | 1.003 (0.998, 1.009) | 0.222 | -0.001 | 0.999 (0.998, 1) | **9.9E-03** |
| | Fractions: Total dose | -0.001 | 0.999 (0.9988, 0.9997) | **0.003** | 0.001 | 1.001 (1, 1.001) | **<0.001** |

categories of causes of death (Table 2, tumor p = 0.001, lymphoma p<0.001, non-tumor p <0.001, CDU p <0.001). Sex was also significant in all models. Males had a higher death hazard for tumors, while females were at greater risk for all other causes of death, including lymphomas. Additionally, we examined the top two causes of death specifically—lung tumors and generalized non-thymic lymphomas (S5 Fig in S2 File). Males were only at a greater risk of lung tumor death, while females had a higher hazard ratio for tumors excluding lung tumors and generalized non-thymic lymphomas.

## Cumulative incidence rates in deaths from lymphomas, tumors, non-tumors, and causes of death unknown in gamma irradiated mice varied greatly for each COD based on total dose and fractionation status

Fig 4A shows the non-parametric cumulative incidence of death for each of the main causes of death. Lymphomas were the most prevalent COD, followed by lung tumors, tumors (excluding lung tumors), non-tumors, and CDU. When we divided the data into control mice (Fig 4B)

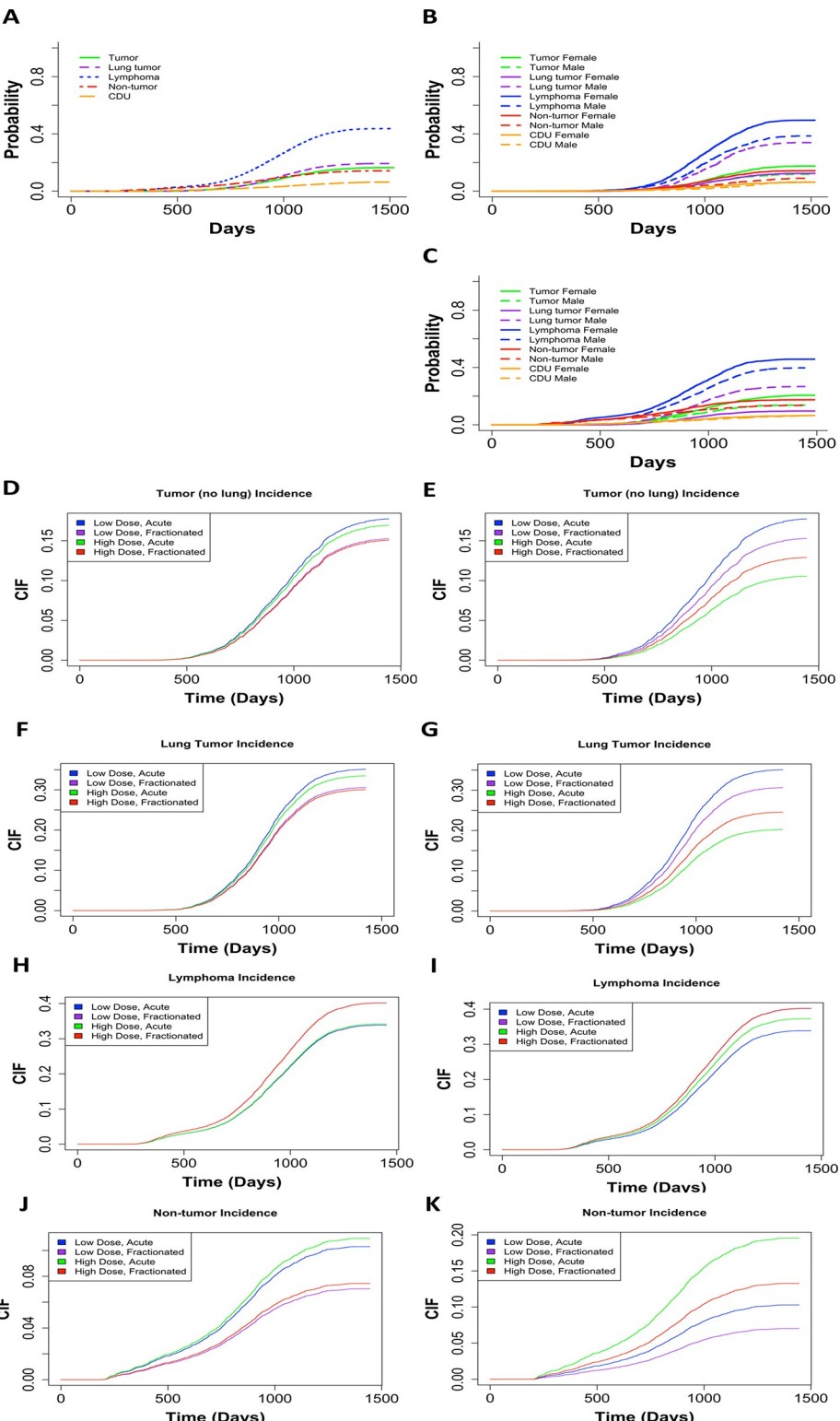

**Fig 4. (A)** Non-parametric CIF for the 5 main categories of COD without grouping. Non-parametric CIF for the 5 main categories of COD grouped my sex for control/sham irradiated mice **(B)** or gamma irradiated mice **(C)**. P-values for differences in sex for each COD for mice graphed in (B): Tumor = 2.39E-06, lung tumor = 0, lymphoma = 1.99E-12, non-tumor = 5.04E-7, CDU 0.965, and for mice graphed in (C): Tumor = 3.97E-12, lung tumor = 0, lymphoma = 2.57E-08, non-tumor = 1.54E-05, CDU = 0.60. Predicted outcome under the following conditions: low

dose = 0.1Gy, high dose = 1Gy for (**D**) tumors (excluding lung), (**F**) lung tumors, (**H**) lymphomas, and (**J**) non-tumors. Predicted outcome under the following conditions: low dose = 0.1Gy, high dose = 10Gy for (**E**) tumors (excluding lung), (**G**) lung tumors, (**I**) lymphomas, and (**K**) non-tumors. All predicted outputs are under the following conditions: Sex = males, acute = 1 fraction, fractionated = 60 fractions. Model output with parameter estimates, hazard ratios (95% confidence interval), and p-value are listed in **Table 2**.

and gamma irradiated mice groups (Fig 4C), the first instances of death were observed around 750 days in control mice and around 250 days in gamma irradiated animals. These graphs were also subdivided by gender. For control and gamma irradiated mice, males had a lower incidence of lymphoma, tumors (excluding lung tumors), and non-tumors COD, while females had a much lower incidence of lung tumors. The differences between males and females were significant for both controls and gamma irradiated mice for all causes of death, except CDU (Table 2).

In addition to calculating cause specific hazards for competing risks, it is important to examine subdistribution hazards, also known as cumulative incidence functions, using the Fine and Grey method [32, 33, 41, 42]. The parameter estimates from subdistribution hazards have a less direct interpretation, but instead elucidate the overall probability of a particular outcome. We used the Fine and Grey method controlling for sex, age first irradiated, number of fractions, total dose, and the interaction between fractions and total dose, with competing risk groups as lymphoma, lung tumors, tumors (excluding lung tumors), non-tumors, and CDU. For tumors (excluding lung tumors) females were more susceptible but fractionation and increased dose both decreased tumor incidence (Table 2). By graphing predicted outcomes under varying conditions, we discovered that when the difference between high and low total doses is small (10cGy vs 100cGy), fractionation is the biggest determinate for tumor incidence, with acute exposures resulting in the most tumors (Fig 4D). Conversely, when the high dose was increased to 1000cGy, total dose became the dominant factor and low dose conditions resulted in the most tumor incidences (Fig 4E). Under all conditions, low dose acute exposures resulted in the greatest tumor incidence. Fractionation and dose had the same impact on lung tumor incidence as in all other tumor incidence (Fig 4F and 4G). However, males were more likely than females to die of lung tumors specifically, which matches the cause specific hazards results (Table 2).

Examining lymphoma deaths, females were at a greater risk of death than males and increasing the total dose and number of fractions both increased the risk of death (Table 2). Predicted outcomes showed that with a 10-fold difference between high and low total doses, fractionation was the main determinate for lymphoma incidence and fractionated exposures resulted in the most lymphoma cases (Fig 4H). Predicted outcomes with a 100-fold difference between high and low total doses resulted in total dose becoming the dominant factor and high dose conditions produced the most lymphoma incidences (Fig 4I). For all conditions, high dose fractionated exposures resulted in the greatest lymphoma incidence, which is the exact opposite from the trend observed for tumor deaths.

Non-tumor deaths were more prevalent in female mice compared to male mice, total dose increased the probability of non-tumor deaths, and fractionation decreased the probability of a non-tumor death (Table 2). When we examined the predicted outcome using a 10-fold difference between high and low total doses, fractionation had the greatest impact on lymphoma incidence, with acute exposures resulting in the most non-tumor cases (Fig 4J). We observed that dose became the dominant factor when we assessed a 100-fold difference between high and low total doses, and high dose conditions resulted in the most non-tumor deaths (Fig 4K). High dose acute exposures resulted in the greatest non-tumor incidence consistently for all conditions we analyzed.

## Lymphoma and non-tumors both had higher incidence rates with high doses at earlier times than previously anticipated

The CIF regression models for death by lymphomas and non-tumors both demonstrated a shoulder along the CIF curve around 250–500 days (Fig 4H–4K). When we filtered out total doses above 6Gy (S6 Fig in S2 File), this shoulder disappeared for lymphoma deaths and non-tumor deaths. Because tumors (excluding lung tumors) and lung tumors (Fig 4D–4G) did not exhibit the same shoulder, these results act as an internal negative control. After filtering out mice exposed to total doses over 6Gy for tumors and lung tumors, the shape of the CIF curves did not change (S6B and S6D Fig in S2 File).

## Janus datasets showed similar results to IES data in male versus female comparisons and dose response trends for causes of death

Several large-scale chronic exposure studies were done at IES using both genders of B6C3F1 mice. Work by Tanaka and others [30] at IES was focused on specific causes of death in response to gamma irradiation and we compared these data with Janus data. As mentioned in the methods, the B6C3F1 mice strain used at IES was genetically similar to the B6C3F1 mice used in the Janus experiments. These F1 mice came from crosses of the same female strain C57BL/6J and two different strains of male mice: C3H/HeJ for IES vs. BALB/cJ for Janus experiments. When examining cause of death between groups (Table 3), we found that hematopoietic system diseases are the most common cause of death for both sets of mice. Additionally, females died of hematopoietic diseases more than males in both datasets. The next most common cause of death for Janus mice was respiratory disease, a result driven by the high incidence of lung cancer in B6CF1 mice from Janus studies. B6C3F1 mice died much less frequently of respiratory disease compared to the B6CF1 mice. However, respiratory diseases were more common in male mice in both datasets. Finally, in male B6C3F1 mice, digestive diseases were much more frequent than in animals used in Janus experiments.

## Discussion

Janus experiments were analyzed in many different ways over the years [14, 17, 20, 23, 24, 26–28, 43, 46–48] and each new approach for analysis of these data brought novel information about the effects of dose fractionation. Common to all these studies is the fact that they either considered each Janus experiment individually, or combined all of them into a single dataset. This is the first study where individual Janus experiments were combined based on control animal datasets compatibility. By analyzing control mice in a way that allowed us to pool Janus experiments together, we gained statistical power to run tests on the importance of fractionation for specific causes of death. The Janus experiments were originally designed with this in mind and taking advantage of the consistency between experiments for a large-scale study was extremely effective. We were able to determine under which circumstances fractionation had a rescuing effect and track changes in risk based on gender for specific causes of death. Our method for pooling data together can be used for future analysis on the Janus dataset using different modeling techniques and answering novel biological questions. Moreover, it is conceivable that a similar approach could be applied to other types of datasets. For example, one can imagine a scenario where animal studies conducted in different laboratories where control animals have similar distribution of cause of death diseases could be combined for a complex combined evaluation of different test conditions.

One of the most interesting findings from the control mice analysis was that mice sham irradiated with 300 fractions died significantly earlier than animals exposed to fewer fractions

**Table 3. The percentage of deaths due to each individual cause of death listed for a comparison between B6CF1 Janus mice and B6C3F1 IES mice.**

| Males | | B6C3F1 IES mice | | | B6CF1 Janus mice | | |
|---|---|---|---|---|---|---|---|
| | | 0Gy | 0.4Gy | 8Gy | 0Gy | 9.2Gy | 9.6Gy |
| | | | 400 fractions | 400 fractions | Sham fractions | 24 fractions | 120 fractions |
| | | | 22h/day | 22h/day | | 45min/fraction | 45 min/fraction |
| | | | 1.1mGy/day | 21mGy/day | | 0.85cGy/min | 0.006 cGy/min |
| | COD | | | | | | |
| | Circulatory System | 8.20% | 9.20% | 13.40% | 5.70% | 8.40% | 6.90% |
| | Digestive System | 24.70% | 27.80% | 19.40% | 3.40% | 2.10% | 0.00% |
| | Endocrine System | 0.60% | 0.40% | 0.20% | 0.30% | 0.50% | 0.00% |
| | Hematopoietic System | 40.60% | 40.40% | 41.50% | 38.80% | 29.50% | 40.30% |
| | Male Reproductive System | 0.00% | 0.40% | 0.00% | 0.40% | 1.10% | 0.00% |
| | Nervous System | 0.40% | 0.00% | 0.20% | 0.10% | 0.00% | 0.00% |
| | Nonneoplastic | 11.40% | 8.80% | 9.20% | 9.10% | 22.60% | 16.70% |
| | Respiratory System | 7.20% | 6.40% | 6.80% | 34.10% | 21.60% | 26.40% |
| | Skeletal System | 0.60% | 0.60% | 0.00% | 0.10% | 0.50% | 0.00% |
| | Skin | 0.60% | 0.00% | 0.40% | 0.00% | 0.00% | 0.00% |
| | Soft Tissue | 5.00% | 4.20% | 5.80% | 1.10% | 1.60% | 0.00% |
| | Special Sense Organs | 0.20% | 0.40% | 1.20% | 0.40% | 0.00% | 0.00% |
| | Unknown | 0.00% | 0.60% | 1.00% | 6.20% | 6.80% | 6.90% |
| | Urinary | 0.00% | 0.40% | 0.40% | 0.20% | 2.60% | 2.80% |
| | Mesothelium/Other Tumor | 0.40% | 0.40% | 0.40% | 0.10% | 0.00% | 0.00% |
| Females | COD | | | | | | |
| | Circulatory System | 3.20% | 3.60% | 5.60% | 4.10% | 5.10% | 7.30% |
| | Digestive System | 2.80% | 2.40% | 3.80% | 1.40% | 2.20% | 2.00% |
| | Endocrine sytem | 5.00% | 4.20% | 2.00% | 1.10% | 2.20% | 1.30% |
| | Female Reproductive System | 4.00% | 2.80% | 6.40% | 4.90% | 6.40% | 7.90% |
| | Hematopoietic System | 63.60% | 59.20% | 57.80% | 49.80% | 39.80% | 31.80% |
| | Nervous System | 0.00% | 0.20% | 0.20% | 0.00% | 0.30% | 0.30% |
| | Nonneoplastic | 9.20% | 12.50% | 7.60% | 14.30% | 17.80% | 20.20% |
| | Respiratory System | 1.20% | 1.60% | 3.00% | 12.70% | 10.80% | 12.30% |
| | Skeletal System | 0.00% | 1.40% | 1.40% | 0.50% | 1.00% | 0.30% |
| | Skin | 0.00% | 0.20% | 0.20% | 0.00% | 0.00% | 0.00% |
| | Soft Tissue | 8.60% | 10.30% | 9.20% | 4.10% | 4.50% | 3.30% |
| | Special Sense Organs | 1.00% | 0.80% | 1.40% | 0.20% | 0.30% | 1.30% |
| | Unknown | 1.40% | 0.80% | 1.20% | 6.20% | 9.20% | 9.60% |
| | Urinary | 0.00% | 0.00% | 0.20% | 0.40% | 0.30% | 2.00% |
| | Mesothelium/Other Tumor | 0.00% | 0.00% | 0.00% | 0.20% | 0.00% | 0.30% |

during their sham irradiations. Mice given 300 sham irradiation fractions died earlier due to tumors and CDU. The simplest and most likely explanation for this phenomenon is general stress caused by frequent exposure to unfamiliar circumstances. Sham exposures involved transporting the mice from the room where they were housed to the room with the irradiator. The irradiator was turned off and there was no excess radiation in the room. It is known that transporting mice induces a stress response [49, 50]. The mice that received 300 sham fractions also had an increase in CDU incidences compared to mice that received fewer fractions. It is possible that the observed decrease in lung tumor and non-tumor deaths was due to misclassification of those deaths as CDU. Further investigation of this mouse cohort may provide us with new insights into the stress response. Analyzing available tissue samples from these mice

could enable us to explore cytological or molecular indicators of stress. These results would not only be beneficial for animal studies in radiation biology, but for any investigators utilizing animals in their research.

In all comparisons between acute and fractionated exposures, fractionation significantly decreased the death hazard in gamma irradiated mice. Moreover, fractionation was equally protective for all four pooled categories of diseases (lymphoma, tumors, non-tumors, and CDU) and specific diseases such as non-thymic lymphoma. Lung tumors were the only causes of death that were not significantly affected by fractionation. B6CF1 males were at a significantly higher risk for lung tumors than females. Interestingly, when we excluded lung tumors and examined all other types of tumors, males were at a lower risk than females for all other causes of death. Heidenreich et al. investigated lung tumors from the Janus datasets using Kaplan-Meier plots and the two-step clonal expansion (TSCE) model [25]. They concluded that males had a higher lung cancer risk than females in control and gamma irradiated mice. They also found that more fractions administered over a longer duration resulted in less lung tumor risk, again agreeing with the results we found using Cox PH.

Examining CIFs to determine the probabilities for death due to distinct diseases under varying conditions produced many intriguing results. Increasing the number of fractions and increasing the total dose both decreased the incidences of tumors and lung tumors specifically. This finding is not surprising because tumors in mice develop more slowly than lymphomas and even more slowly than non-tumors such as radiation induced pneumonitis. Therefore, a mouse exposed to a high dose of gamma rays would most likely die of non-cancer cause of death before a tumor has time to fully advance.

Mice exposed to fractionated irradiation died from lymphoma more frequently when compared to mice that received acute exposures, while mice exposed to acute exposures developed more non-tumors. Non-tumors were the most common cause of death in response to higher doses and acute exposures. Considering that non-irradiated B6CF1 mice begin to develop lymphomatous spleens by 600 to 700 days of age and that even few spontaneous lymphoma cells have the immunosuppressive effect in spleen [51], it is possible that other causes of death may also be partially dependent on pre-symptomatic lymphoma development. Overall, B6CF1 mice are a robust hybrid mouse strain, immunocompetent and long lived (853 ± 10 days on average [51]), and almost as radiation resistant as its more radiation resistant parent strain C57/BL mice ($LD_{50/30}$ of about 6.6 Gy for mice exposed at 120 days of age) [45].

Lymphomas are easily induced in response to ionizing radiation in rodents. It is typically considered a risk associated with lower doses of ionizing radiation compared to non-tumors. However, our results showed an early increase in lymphoma incidences when the dose delivered to animals was above 6 Gy. A 6Gy cutoff was chosen because all of the animals exposed to doses above 6 Gy received fractionated irradiation. This was done because the $LD_{50/30}$ dose for B6CF1 mice is 6.54 or 6.75 Gy for males and females respectively [25] and acute exposures above 6Gy would result in an animal death rate incompatible with robust experimental data. In mice that received total doses over 6Gy (all fractionated exposures), lymphoma deaths began as early as 300 days. The early death shoulder observed in CIF curves for lymphomas is no longer present when excluding data for total doses above 6Gy. Non-tumor deaths demonstrated the same shoulder when the full range of total doses were included to fit the model. Again, the shoulder disappears after removing the data for total doses over 6Gy.

We compared the Janus dataset on fractionated radiation with the IES datasets on chronic irradiation. This comparison between B6CF1 mice used in Janus experiments and B6C3F1 mice used by Tanaka and others [30] showed a high degree of similarity despite different radiation delivery approaches and genetic differences between the two strains. The most pronounced differences between B6CF1 and B6C3F1 mice were associated with male mice, which

could be attributed to these two hybrid strains differing by the paternally contributing mice. While respiratory system complications affected both F1 mouse hybrids discussed here, not all mice have the same association between gender and radiation associated respiratory diseases. For example, RFM mice exposed to x-rays males were at less risk than females for lung tumors, indicating that differential gender susceptibilities to lung tumors are strain specific [52–54]. IES results also showed more death due to digestive diseases, with the effect being most obvious in male mice. Janus mice were not kept in a sterile environment nor fed sterile food. The bacteria present in the guts of Janus mice would have increased the local immune response and may explain the lower percentage of deaths due to digestive diseases. These noticeable differences in digestive and respiratory disease proportions could be explained by several differences in housing. Not only were IES mice specific pathogen free, while Janus mice were not, but it is also likely that standard housing conditions changed between 1972 and 2004. Beyond standard conditions over time, conditions are likely variable across universities and countries, as well. In humans, females have been shown to be at a greater risk of lung cancer than males [55, 56]. Determining the cause for changes in lung tumor sensitivity in response to ionizing radiation between male and female mice could lead to a better understanding of the radiation induction of lung tumors. Given our current, limited amount of information, RFM mice appear to be a better model system for simulating gender differences of humans for lung tumor risk. While radiation doses associated with the $LD_{50/30}$, for example, vary significantly between rodents and humans [6, 45], most interspecies comparisons focus on proportional life shortening [57].

In conclusion, we propose to continue to evaluate the NURA database using different models and applying them to different subsets of data in order to outline the finer nuances of consequences of radiation exposures. The differences we described in radiation exposure outcomes that change with alterations in radiation delivery highlight that biological responses to whole body irradiation most likely cannot be described by a single factor that could be applied for the entire spectrum of possible fractionation scenarios.

## Supporting information

**S1 File.**
(DOCX)

**S2 File.**
(PDF)

## Acknowledgments

The authors would like to thank Edward Malthouse for his input on the statistical methods used and Carissa Ritner and Benjamin Haley for their constant support and thoughtful discussions.

## Author Contributions

**Conceptualization:** Alia Zander, Tatjana Paunesku, Gayle E. Woloschak.

**Formal analysis:** Alia Zander.

**Funding acquisition:** Tatjana Paunesku, Gayle E. Woloschak.

**Investigation:** Alia Zander, Tatjana Paunesku, Gayle E. Woloschak.

**Methodology:** Alia Zander, Tatjana Paunesku, Gayle E. Woloschak.

**Project administration:** Tatjana Paunesku, Gayle E. Woloschak.

**Resources:** Tatjana Paunesku, Gayle E. Woloschak.

**Supervision:** Tatjana Paunesku, Gayle E. Woloschak.

**Validation:** Alia Zander.

**Visualization:** Alia Zander.

**Writing – original draft:** Alia Zander.

**Writing – review & editing:** Alia Zander, Tatjana Paunesku, Gayle E. Woloschak.

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
