## [Decision Letter · Decision Letter 0]

4 May 2020

PONE-D-20-07144

Analyses of cancer incidence and other morbidities in gamma irradiated B6CF1 mice

PLOS ONE

Dear Dr. Woloschak,

Thank you for submitting your manuscript to PLOS ONE. After careful consideration, we feel that it has merit but does not fully meet PLOS ONE’s publication criteria as it currently stands. Therefore, we invite you to submit a revised version of the manuscript that addresses the points raised during the review process.

We would appreciate receiving your revised manuscript by Jun 18 2020 11:59PM. To enhance the reproducibility of your results, we recommend that if applicable you deposit your laboratory protocols in protocols.io, where a protocol can be assigned its own identifier (DOI) such that it can be cited independently in the future. For instructions see: http://journals.plos.org/plosone/s/submission-guidelines#loc-laboratory-protocols

We look forward to receiving your revised manuscript.

Kind regards,

Jian Jian Li, M.D., Ph.D.

Academic Editor

PLOS ONE

Journal Requirements:

2. Thank you for stating the following in the Competing Interests Section of the submission form:"The authors have declared that no competing interests exist."

Please update your Competing Interests Statement to confirm that author Gayle E. Woloschak is a section editor for PLOS ONE.

Additional Editor Comments (if provided):

This work contains some important information on radiation-associated carcinogenic risk. Due to the huge number of animals involved, it is suggested that to recheck the statistical approaches used for validating the data in this study.

Reviewers' comments:

Reviewer's Responses to Questions

**Comments to the Author**

1. Is the manuscript technically sound, and do the data support the conclusions?

Reviewer #1: Yes

Reviewer #2: Yes

2. Has the statistical analysis been performed appropriately and rigorously? 

Reviewer #1: Yes

Reviewer #2: Yes

3. Have the authors made all data underlying the findings in their manuscript fully available?

Reviewer #1: Yes

Reviewer #2: Yes

4. Is the manuscript presented in an intelligible fashion and written in standard English?

Reviewer #1: Yes

Reviewer #2: Yes

5. Review Comments to the Author

Reviewer #1: This article studied the risks associated with ionizing radiation by analyzing the Northwestern University Radiation Archive from a series of 10 individual neutron and gamma irradiation experiments conducted on over 50,000 mice. They used rigorous statistical testing on control mice from all Janus experiments to select studies that could be compared to differences among the controls as well as experimental animals. Experiments are conducted rigorously, with appropriate controls, replication, and sample sizes. The conclusions are drawn appropriately based on the data presented. . The manuscript was presented in an intelligible fashion and written in standard English.

There are some comments:

1.Janus experiments were analyzed in many different ways over the years, what are the difference of these methods, and what is the meaning of the new approach for analysis used in this article?

2.The purpose and of this article should be stated in the discussion.

3.How radiation doses are converted between humans and mice?

Reviewer #2: This paper employs the NURA database of mice receiving irradiation treatment to analyze the impact of irradiation dose, fractions, sex on the survival and incidence of death due to different causes, including lymphoma, tumors except lung cancer, lung cancer and CDU, which is designed and conducted logically and seriously. It shows some information for researchers and radiation oncologists, especially the difference of fraction and total dose in the influcence of tumor incidence.

1. Mice sham irradiated with 300 fractions die significantly ealier than those mice with fewer fractions. Does the truly irradiated group have a similar finding?

2. The genetic and immune background of B6CF1 mice should be discussed in the discussion.

3. Fig1,2 and 3, the statistic significance should be caculated and labeled both in the results part and figures.

4. In Fig 1, It seems there was a sinificant difference between two groups in tumors(excluding lung), CDU and lymphoma. And possibilities that CDU may affect the death caused by tumors should be discussed.

5. Table 1 legend, Parameter estimates, hazard ratios with 95% confidence interval, and p-values for main

Cox Proportional Hazards model in Fig 1D. It should be Fig1D ? or Fig 2D?

6. since the sex difference is shared in common by control and irradiation, the sex role in death affected by irradiation should be adjusted.

7. In Line 392, the B6C3F1 mice strain used at IES was genetically similar to the B6C3F1 mice used in the Janus experiments. It should be B6CF1 or B6C3F1 in the Janus?

8. The indicated clinical impact and potential use of the findings in this study are recommended to be discussed.

9. Its not clear whether neutron irradiation treatment is excluded or not.

10. all figures in the integrated muanscript is too vague to see. The orignal figures in tiff format are good.

6. PLOS authors have the option to publish the peer review history of their article (what does this mean?). If published, this will include your full peer review and any attached files.

Reviewer #1: No

Reviewer #2: No

---

## [Author Response · Author response to Decision Letter 0]

17 Jun 2020

Dear Reviews,

Thank you for taking the time to critically read through our manuscript and give helpful feedback. We responded your comments/suggestions to the best of our ability and details can be found below; we indicated added text by underlining it:

Reviewer 1:

1. Janus experiments were analyzed in many different ways over the years, what are the difference of these methods, and what is the meaning of the new approach for analysis used in this article?

Recently, we have used Janus archive to re-evaluate dose and dose rate effectiveness factor for life shortening modulation due to fractionation (Haley et al, 2015) and in this work we recreated formalism developed in BEIR VII as R script that could be shared and verified by the community. Having done so, we felt that it would be equally valuable to use R scripts developed for clinical studies for a re-evaluation of animal specific causes of death, both to explore the new approach to analysis (use and exchange of R scripts in github is a recent development) and to evaluate how suitable these tools would be if we wished to look into disease-specific DDREF evaluation. While we continue to work in this direction, this manuscript covers a segment of our effort that has to do with optimization of combining different experiments for joint study. We have now tried to explain this better:

1) in the abstract: “This study systematically cross-compared outcomes of different modes of fractionation evaluated across different Janus experiments and a wide span of total doses.”

2) in the introduction: “Numerous studies used the NURA (also known as Janus) database. In most cases, different Janus experiments were used separately (17, 20, 23-28) or else combined all together into a single dataset (14). In this study, however, many but not all Janus experiments were combined into a dataset – process of selection was based on comparability of control animal datasets from sham irradiation conditions in different Janus experiments.”

3) in the methods: “…; different species such as Peromyscus leucopus (white-footed deer mouse) were excluded from this study because of the species to species differences between the controls and in response to radiation” 

and later, in the methods subsection discussing IES vs. Janus data:

“The differences in disease incidence between the control animals point out that only some of these disease “endpoints” are appropriate for direct comparisons between strains when different “test conditions” are being evaluated. “

4) in the discussion: “Common to all these studies is the fact that they either considered each Janus experiment individually, or combined all of them into a single dataset. This is the first study where individual Janus experiments were combined based on control animal datasets compatibility…… Moreover, it is conceivable that a similar approach could be applied for other types of datasets. For example, one can imagine a scenario where animal studies conducted in different laboratories where control animals have similar distribution of cause of death diseases could be combined for a complex combined evaluation of different test conditions.”

2. The purpose and of this article should be stated in the discussion.

 Please see the response to question 1 from reviewer 1.

3. How radiation doses are converted between humans and mice?

We have now added the following to the discussion:

“In addition, it should be noted that radiation doses associated with LD50/30, for example, very significantly in rodents and humans (6, 45), and most interspecies comparisons focus on proportional life shortening (56).”

Reviewer 2:

1. Mice sham irradiated with 300 fractions die significantly ealier than those mice with fewer fractions. Does the truly irradiated group have a similar finding?

Mice that received radiation in 300 fractions showed a decrease in death hazard compared to acutely exposed mice. We excluded them from the main study because the sham 300-fractions controls died significantly earlier indicating that the data on mice with 300 fractions could no longer be pooled together. To make this a bit clearer we have included:

1) in the abstract: “For controls, mice sham irradiated with 300 fractions died significantly earlier than those with fewer sham fractions and were excluded from pooled control dataset.”

2) in the results: Supp Fig 3 I and J show the main cox PH model including the mice irradiated in 300 fractions with fractions treated as a continuous variable. Supp Fig 3 K and L show the main cox PH model including the mice irradiated 300 fractions treated as a categorical variable. For both models, the interaction term shows that mice have a lower death hazard if they receive their total dose in 300 fractions compared to acute exposures. The specific text is: “Notably, gamma irradiated mice that received their total doses in 300 fractions had a decrease in the death hazard compared to mice that received acute exposures, even with the added stress that caused control mice to die significantly earlier (S3 Fig I-L).”

3) in the discussion: “The mice that received 300 sham fractions also had an increase in CDU incidences compared to mice that received fewer fractions. It is possible that the observed decrease in lung tumor and non-tumor deaths was due to misclassification of those deaths as CDU.”

2. The genetic and immune background of B6CF1 mice should be discussed in the discussion.

Thank you for this suggestion. We added more to the discussion: “Considering that non-irradiated B6CF1 mice begin to develop lymphomatous spleens by 600 to 700 days of age and that even few spontaneous lymphoma cells have the immunosuppressive effect in spleen (51), it is possible that other causes of death may also be partially dependent on pre-symptomatic lymphoma development. Overall, B6CF1 mice are a robust hybrid mouse strain, immunocompetent and long lived (853 ± 10 days on average (51)), and almost as radiation resistant as its more radiation resistant parent strain C57/BL mice (LD50/30 of about 6.6 Gy for mice exposed at 120 days of age) (45).”

3. Fig1,2 and 3, the statistic significance should be caculated and labeled both in the results part and figures.

Figure 1 – the statistical significance in Fig 1 A was previously listed in Supp Table 6, but we moved it to be Fig 1B, next to the figure 1A and added the p-value to the figure legend and results section. We added statistical significance for the rest of the figures to the results section as well.

Figure 2 – the best statistical test for these values come from the coxph models. We added the significance output from the cox ph model to the text in the results section and figure legend. 

Figure 3 – we added the p-values for the interaction term between fractions and total dose to the text in the results section and the figure legend. 

4. In Fig 1, It seems there was a sinificant difference between two groups in tumors(excluding lung), CDU and lymphoma. And possibilities that CDU may affect the death caused by tumors should be discussed.

This is an excellent point and we added text to the discussion: “The mice that received 300 sham fractions also had an increase in CDU incidences compared to mice that received fewer fractions. It is possible that the observed decrease in lung tumor and non-tumor deaths was due to misclassification of those deaths as CDU.” (also please see reviewer2:1)

5. Table 1 legend, Parameter estimates, hazard ratios with 95% confidence interval, and p-values for main Cox Proportional Hazards model in Fig 1D. It should be Fig1D ? or Fig 2D?

Thank you for noticing this. We updated the table to say Fig 2D.

6. since the sex difference is shared in common by control and irradiation, the sex role in death affected by irradiation should be adjusted.

Thank you for pointing out that because baseline differences exist between sexes, we cannot tell which group is more sensitive to radiation treatment. We added an analysis that includes the interaction term between sex and total dose in an effort to see how each gender responds to increased ionizing radiation exposure. We found that the baseline differences become significantly more dramatic as total dose increases. These new results (graph and table with model output) are in supplemental figure 3 M-N with other robustness tests.

We have added text to results:

“When adding a new interaction term between sex and total dose, we found that as the total dose increased, the decreased death hazard in males was more pronounced (S3 Fig M-N).”

7. In Line 392, the B6C3F1 mice strain used at IES was genetically similar to the B6C3F1 mice used in the Janus experiments. It should be B6CF1 or B6C3F1 in the Janus?

The mice used in the Janus experiments were B6CF1 mice, while the mice from IES were B6C3F1 mice. We have modified and/or added text to clarify:

1) in methods: “Studies at IES involved chronic low dose rate gamma irradiations of specific-pathogen free (SPF) B6C3F1 mice, F1 progeny of C57BL/6J females (B6) and C3H/HeJ males. The B6CF1 mice, F1 progeny of C57BL/6J females (B6) and BALB/cJ males, were used during the Janus experiments. Both strains are F1 hybrids that share the same maternal strain C57BL/6.”

2) in results: “These F1 mice came from crosses of the same female strain C57BL/6J and two different strains of male mice: C3H/HeJ for IES vs. BALB/cJ for Janus experiments.”

8. The indicated clinical impact and potential use of the findings in this study are recommended to be discussed.

In response to reviewer1:1 & 2 and this comment, we have added the following sentences to the discussion:

“Common to all these studies is the fact that they either considered each Janus experiment individually, or combined all of them into a single dataset. This is the first study where individual Janus experiments were combined based on control animal datasets compatibility…… Moreover, it is conceivable that a similar approach could be applied for other types of datasets. For example, one can imagine a scenario where animal studies conducted in different laboratories where control animals have similar distribution of cause of death diseases could be combined for a complex combined evaluation of different test conditions.”

9. Its not clear whether neutron irradiation treatment is excluded or not.

Neutron irradiated mice were not included in this analysis, but are instead included in another publication that is under review at PLOS ONE. To make this more obvious to readers we added new text in the introduction: “We examined whether fractionation, age at which a mouse was first irradiated, and gender modulated the overall death hazard and frequency for specific causes of death in gamma irradiated mice.”

in the methods section: “For this analysis, we focused on gamma irradiated mice. Neutron irradiated mice were studied in a separate analysis.”

10. all figures in the integrated muanscript is too vague to see. The orignal figures in tiff format are good.

We anticipate that the journal will handle this issue or help us if there is need for more work from our end. Thank you for drawing this to everyone’s attention.

---

## [Decision Letter · Decision Letter 1]

22 Jul 2020

Analyses of cancer incidence and other morbidities in gamma irradiated B6CF1 mice

PONE-D-20-07144R1

Dear Dr. Gayle E. Woloschak,

We’re pleased to inform you that your manuscript has been judged scientifically suitable for publication and will be formally accepted for publication once it meets all outstanding technical requirements.

Kind regards,

Roberto Amendola, Ph.D

Academic Editor

PLOS ONE

Additional Editor Comments (optional):

Reviewers' comments:

Reviewer's Responses to Questions

**Comments to the Author**

1. If the authors have adequately addressed your comments raised in a previous round of review and you feel that this manuscript is now acceptable for publication, you may indicate that here to bypass the “Comments to the Author” section, enter your conflict of interest statement in the “Confidential to Editor” section, and submit your "Accept" recommendation.

Reviewer #2: All comments have been addressed

2. Is the manuscript technically sound, and do the data support the conclusions?

Reviewer #2: Yes

3. Has the statistical analysis been performed appropriately and rigorously? 

Reviewer #2: Yes

4. Have the authors made all data underlying the findings in their manuscript fully available?

Reviewer #2: Yes

5. Is the manuscript presented in an intelligible fashion and written in standard English?

Reviewer #2: Yes

6. Review Comments to the Author

Reviewer #2: The authors clearly answered the questions and supporting experiments were supplemented. I believe this paper is ready to be published.

7. PLOS authors have the option to publish the peer review history of their article (what does this mean?). If published, this will include your full peer review and any attached files.

Reviewer #2: No

---

## [Editor Report · Acceptance letter]

30 Jul 2020

PONE-D-20-07144R1 

Analyses of cancer incidence and other morbidities in gamma irradiated B6CF1 mice 

Dear Dr. Woloschak:

I'm pleased to inform you that your manuscript has been deemed suitable for publication in PLOS ONE. Congratulations! Your manuscript is now with our production department. 

Kind regards, 

on behalf of

Dr. Roberto Amendola 

Academic Editor

PLOS ONE